# A High-Precision Water Body Extraction Method Based on Improved Lightweight U-Net

Shihao An  and Xiaoping Rui *

School of Earth Science and Engineering, Hohai University, Nanjing 211100, China
* Correspondence: ruixp@hhu.edu.cn

**Abstract:** The rapid, accurate extraction of water body information is critical for water resource management and disaster assessment. Its data foundation was mostly provided by remote sensing images through deep learning methods. However, the methods still require the improvement of recognition accuracy and reduction of model size. As a solution, this paper proposed a new high-precision convolutional neural network for water body extraction. This network's structural design is based on the assumption that the extraction effect of a convolutional neural network is independent from its parameters number, thus the recognition effect could be effectively improved through reasonable adjustment of the network structure according to characteristics of water bodies on high-resolution remote sensing images. It brings two critical improvements. Firstly, the number of downsampling layers was reduced to adapt to the low resolution of remote sensing imagery. Secondly, the bottleneck structure has also been updated to fit the decoder–encoder framework. The improved bottleneck structures were nested to ensure the transmission of water characteristics information in the model. In comparison with the other five commonly used networks, the new network has achieved the best results (average overall accuracy: 98.31%, parameter benefit value: 0.2625), indicating the extremely high practical value of this approach.

**Keywords:** water body extraction; GF-2; convolutional neural networks; deep learning

## 1. Introduction

As the most widely distributed substance in nature, water comprises an essential part of terrestrial ecosystems. As the dynamic changes of water bodies can exert a significant impact on human life [1], the rapid and accurate extraction of water body information has been a constant research hotspot. In this context, the development of satellite remote sensing technology has made large-scale dynamic monitoring of water bodies possible, and the inherent rapidity and efficiency of this approach have made water body extraction based on remote sensing images the mainstream method of water body monitoring.

At present, both domestic and foreign scholars have proposed many methods to extract water body information using remote sensing technology [2], such as single-band thresholding [3–5], a multi-band spectral relationship method [6], water indexes [7–10], and remote sensing image classification [11–13], among others. Furthermore, researchers have widely studied and applied the water index since the 1990s because of its simplicity and ability to suppress the background features effectively and highlight the characteristics of water bodies [10]. Inspired by the normalized difference vegetation index (NDVI), McFeeters proposed the normalized difference water index (NDWI) [7]. Unfortunately, although NDWI enhances water area contrast, the method also amplifies the effects of soil and buildings. In response, Xu largely solved this problem with the modified normalized difference water index (MNDWI), which replaced the NIR band with the MIR band [8]. However, because MNDWI uses the mid-infrared band, the range of application for Xu's method is greatly limited, as some remote sensing satellites, such as SPOT6, IKONOS, GF-1 and GF-2, do not carry sensors in the mid-infrared band. Generally, traditional water body

extraction methods rely heavily on hand-selected features, requiring users to have a large amount of professional knowledge. Furthermore, their applicability to different images is very low [14].

In the 2012 ImageNet competition, Krizhevskv's AlexNet significantly reduced the error rate of image recognition from 25.8% to 15.3% [15]. Since that time, deep learning can automatically obtain the feature information of objects from massive training data and provide analysis through these weight coefficients [16]. Such characteristics also make this approach a key technology for the automatic semantic segmentation of remote sensing images. In recent years, deep learning has provided an effective framework for the classification and identification of massive remote sensing image data, and has promoted the gradual development of remote sensing image processing. In 2017, Isikdogan applied deep learning technology to water body identification in remote sensing images, and the accuracy of the DeepWaterMap network that he constructed far exceeded the performance of the MNDWI and Multi-Layer Perceptron (MLP) [17]. In 2018, Li presented the DeepUNet, which was based on the U-Net, and further improved the accuracy of the convolutional neural networks for water body recognition in remote sensing images [18]. Chen explored a water body extraction method that combined an adaptive clustering algorithm and a convolutional neural network. The researcher then applied the method to the classification of high-resolution remote sensing images, obtaining an overall accuracy of up to 99.14% [1]. In 2020, Wang built a system that could automatically acquire and train model data using Google Earth Engine (GEE) and a multiscale convolutional neural network (MSCNN), which greatly improved the automation of water body extraction [19]. In 2021, Li constructed the dense-local-feature-compression (DLFC) network that could adapt to various remote sensing image data [14], which fully demonstrated that the features extracted by convolutional neural networks are universal. According to the focus of these previous studies, the core of using deep learning technology to solve water body recognition in remote sensing images reflects various efforts to build a suitable network structure.

Our analysis of the existing literature shows that most scholars currently focus their research on improving the accuracy of remote sensing imagery water extraction networks. By increasing the model parameters, the deep learning model can obtain more subtle extraction capabilities and it is also a widely used method to improve the accuracy of network extraction. The increase of model parameters means that it takes more space and a longer time to configure the model, which is a problem that has been ignored by scholars who pursue high precision. However, the question arises as to whether high precision and few parameters can never coexist. Zhou, the creator of UNet++ [20], believes that this is not the case, citing such examples as PSPNet [21] and FC-DenseNets [22] to prove the point. While the number of layers for downsampling in the two networks mentioned is one and three, the results surpass those of other deeper networks. In addition, networks with complex structures often require larger storage space and training samples, and network transmission consumes much bandwidth and time, which greatly limits the application of these algorithms on smart terminals. Finally, compared with other images, remote sensing images offer unique characteristics, including lower resolution and much bigger size. As a result, the effect obtained by directly applying the existing network structure model to remote sensing image semantic segmentation is not necessarily optimal.

With these thoughts in mind, we aim to build a network model that would be both lightweight and suitable for remote sensing imagery. Five networks that currently enjoy wide use in image recognition experiments were chosen, including SegNet [23], U-Net [24], ResNet [25], DenseNet [26], and PSPNet [21]. Although SegNet achieved relatively good classification results, an interesting phenomenon was found: though its structure was quite similar to SegNet, U-Net performed poorly in this experiment. These results sparked questions regarding why a network with more convolutional layers than SegNet did not perform as well in terms of classification effect. A contemplation of Zhou et al.'s investigation provided some insight into this apparent contradiction [20]. In order to test the performance of U-Net at various depths, they trained U-Net $L^1$, U-Net $L^2$, U-Net $L^3$

and U-Net $L^4$ with the same medical image sample set and tested their classification effects. According to the test results, U-Net $L^3$ and U-Net $L^4$ demonstrated a very close performance in classification, but U-Net $L^3$ surpassed U-Net $L^4$ in the prediction accuracy of some organs and tissues. Notably, medical imaging and remote sensing imagery share the similarity of fewer data samples. Thus, using a network with many parameters when training the model can preclude the ability to apply some structural effects normally because the number of samples provided by the training does not provide sufficient support to the network to solve the optimal parameters. In addition, the pixel resolution of remote sensing images is much smaller than that of general photographed images. Consequently, the most suitable four-layer downsampling structure in general photographed images is not suited to remote sensing images. In our investigation, U-Net $L^3$ was tested on the same remote sensing dataset. According to our findings, the classification effect of U-Net $L^3$ was far better than that of U-Net $L^4$, which was very consistent with our assumptions. Thus, it could be seen that the traditional image semantic segmentation framework was not necessarily suitable for remote sensing images. To this end, the long-standing four-layer downsampling structure in U-Net was abandoned, switching to a three-layer downsampling structure, and cutting some difficult-to-train parameters. In ResNet50, the author uses the bottleneck structure to replace the two convolutional layers in ResNet34. Some of our ideas come from ResNet50. Compared with convolution kernels of other sizes, this structure can greatly reduce the computational complexity and use more space to expand the depth of networks. From another point of view, if the bottleneck structure simplifies those parameters that are difficult to train, then a model structure with a very good prediction effect can be constructed while ensuring that the computational complexity is small enough. Based on this assumption, the convolutional layer in U-Net $L^3$ was replaced with the bottleneck structure. Reflecting the critical role of the bottleneck and U-Net modules in this network, we named this network BU-Net. For our dataset, GID was used. This is a remote sensing image dataset [27] produced by the State Key Laboratory at Wuhan University. After simple processing, 20,000 training samples were generated for experimental use.

The rest of this article is organized as follows. The second section provides details about the proposed method. Related data, data processing and related experimental results can be found in the third section. Finally, the fourth section offers related discussions, followed by the fifth section, which contains our conclusions.

## 2. Materials and Methods

### 2.1. Improved Network Structure

#### 2.1.1. Architecture of the Proposed Network

Since encoder–decoder represents the most widely used architecture in the field of image semantic segmentation, this classical architecture was used in our network. In the encoder part, we designed three downsampling processes that combined DownBottleneck and max pooling operations to ensure the receptive field and the richness of feature information in different dimensions. In the decoder part, we designed three upsampling processes that combined UpBottleneck and upsampling operations to gradually restore the spatial dimension and detail information. Subsequently, shortcut connections were included between the encoder part and the decoder part for channel connection to facilitate obtaining more feature information in the original image during prediction, which greatly improved the accuracy of the decoder module during decoding. Finally, we used a $1 \times 1$ convolution layer with two filters and a softmax activation function for prediction to obtain a binary image of the same size as the original remote sensing image. Figure 1 displays an illustration of the overall framework.

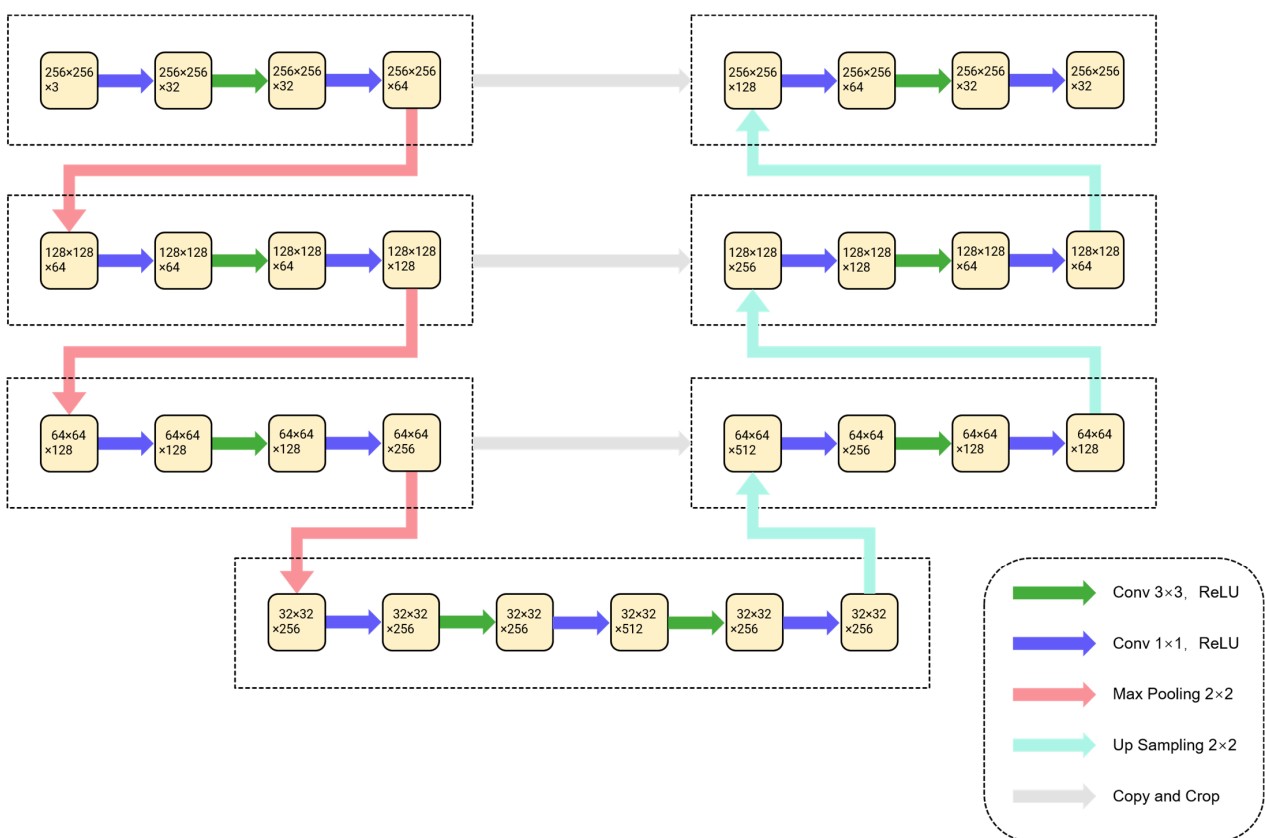

**Figure 1.** Overall framework of BU-Net.

### 2.1.2. Architecture of the Proposed Network

For the processing of the convolution layer, we adopted the construction idea of Resnet50: replace the frequently occurring $3 \times 3$ convolution operation with the bottleneck structure. Taking 64-dimensional input data as an example, the number of parameters required for two consecutive $3 \times 3$ convolution operations with 64 filters was:

$$(3 \times 3 \times 64 \times 64) \times 2 = 73,728. \tag{1}$$

If it was a bottleneck structure, it only needed:

$$1 \times 1 \times 64 \times 64 + 3 \times 3 \times 64 \times 64 + 1 \times 1 \times 64 \times 256 = 57,344, \tag{2}$$

which directly simplified the calculation amount to 78% of the original.

Since the original application field of ResNet was image classification, when building the model, the authors of ResNet paid attention to more deep semantic information, always using a $1 \times 1$ convolution layer with a large number of filters (in most cases, four times the base number) at the end of the bottleneck structure to recombine these high-level dimensional features. However, shallow semantic information was also important in semantic segmentation. When 64-dimensional information data were directly transformed into 256-dimensional deep semantic information after passing through the bottleneck structure, the intermediate process was ignored, leading to some degree of a semantic gap. Therefore, the number of filters in the $1 \times 1$ convolution layer at the end of the bottleneck structure was changed. This alteration allowed images to be processed from low-dimensional information reorganization to high-dimensional information extraction to high-dimensional information integration. We also compared and tested the classification effect of the bottleneck structure under four parameter settings. The result analysis is detailed below.

### 2.1.3. DownBottleneck and UpBottleneck

Since the bottleneck structure was not originally designed for semantic segmentation, only the flow of low-semantic information to high-semantic information was considered in the data extraction. In the decoder part of the encoder–decoder process, it decodes and restores low-semantic information according to high-semantic information. However, it was not feasible to use the bottleneck structure directly. To this end, a module corresponding to the bottleneck structure was designed to realize the flow of high-semantic information to low-semantic information in the decoder part, naming it UpBottleneck (see Figure 2). In order to correspond to the UpBottleneck structure, we also renamed the bottleneck structure in the encoder part to DownBottleneck. The two structures were designed to work together to provide the entire process of remote sensing images, from encoding to decoding.

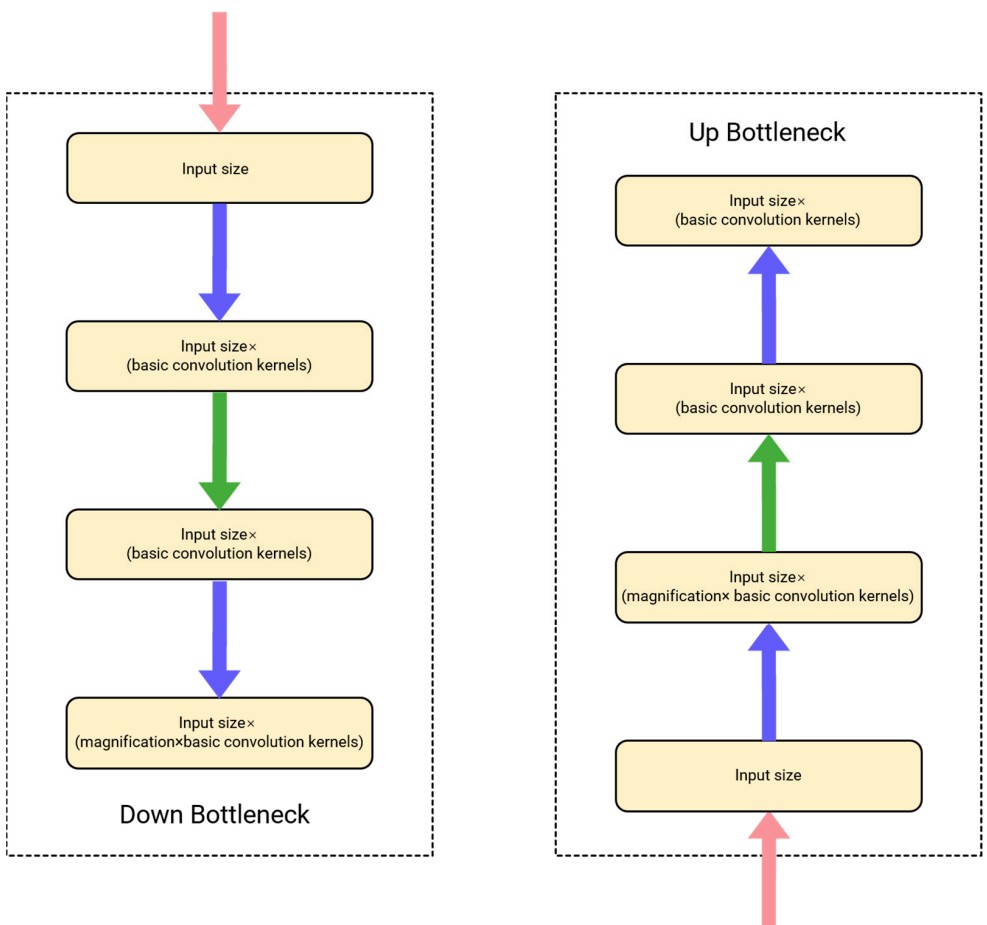

**Figure 2.** Architecture of the DownBottleneck module and the UpBottleneck module.

### 2.1.4. Implementation Details of the Network

With the purpose of ensuring the best extraction effect, setting the correct parameters are necessary. Among them, the activation function used by the network was Linear rectification function (ReLU). Furthermore, a BatchNormalization operation was performed after each convolution operation. In light of the correspondence of the encoder–decoder architecture, the size of upsampling and pooling were both $2 \times 2$, and the stride was 1, in which the pooling operation adopted max pooling. We used cross-entropy as the loss function, with a softmax activation function for result prediction. Additionally, the epoch was set to 100, and the learning rate was set to $1 \times 10^{-4}$. Finally, regarding the optimizer, the Adam optimizer was chosen to speed up the network training process.

### 2.2. Comparison Method

The effect of BU-Net could be verified by being compared with five deep learning networks (U-Net, SegNet, ResNet, DenseNet and PSPNet) in terms of prediction accuracy and model size. Among them, U-Net was originally proposed to solve the problem of medical image segmentation. With its simple and rapid characteristics, it has become one of the most widely used networks in the field of image semantic segmentation. SegNet is an intelligent semantic pixel-wise segmentation project developed by the University of Cambridge, which performs very well in the object recognition task of driverless cars. ResNet has remained an enduring network structure since it was first proposed. Many image classification models are based on this approach, and it is a well-recognized cornerstone structure in the field of computer vision. DenseNet is another genius work that followed ResNet. Compared with ResNet's shortcuts, DenseNet uses more continuous dense connections, from which its name was derived. PSPNet is a lightweight network based on the pyramid pooling module. By employing pooling layers of different sizes, the network can obtain receptive fields of various sizes as a whole and improve overall prediction accuracy with only a few parameters.

### 2.3. Evaluation Metrics

Aiming at the quantification of the prediction accuracy, some commonly applied accuracy metrics were used to evaluate the results. Before introducing the indicators, we first introduced four basic concepts in the binary classification: True Positive (TP), the positive class with accurate classification; False Positive (FP), a negative class misclassified as a positive class; True Negative (TN), the negative class with accurate classification and False Negative (FN), a positive class misclassified as a negative class. Accuracy is used to represent the ratio of the number of correctly classified pixels to the number of all pixels. Accuracy can satisfactorily reflect the classification accuracy. However, when the positive and negative distribution of the sample is extremely unbalanced, its value is greatly affected by more categories of pixels, meaning that it cannot effectively characterize overall recognition accuracy. Precision refers to the ratio of the number of samples accurately classified as positive to all samples classified as positive, while recall signifies the ratio of the number of samples classified as positive to the number of samples in the test dataset. Overall accuracy (OA) represents the overall accuracy, it does not consider the category, only shows the classification of all samples. It is wanted that precision and recall be very high at the same time, but these two metrics could not be raised at the same time, which drove the strategy to pursue a balance between the two: the F1-score. Lastly, Intersection-over-Union (IoU) denotes the ratio of the intersection and union of actual class samples and predicted class samples. The above indicators were calculated according to the following formulas:

$$\text{Accuracy} = \frac{\text{TP} + \text{TN}}{\text{TP} + \text{TN} + \text{FP} + \text{FN}} \tag{3}$$

$$\text{Precision} = \frac{\text{TP}}{\text{TP} + \text{FP}} \tag{4}$$

$$\text{Recall} = \frac{\text{TP}}{\text{TP} + \text{FN}} \tag{5}$$

$$\text{IoU} = \frac{\text{TP}}{\text{TP} + \text{FN} + \text{FP}} \tag{6}$$

$$\text{OA} = \frac{\text{TP} + \text{TN}}{\text{TP} + \text{TN} + \text{FP} + \text{FN}} \tag{7}$$

$$\text{F1} - \text{score} = 2 \times \frac{\text{Precision} \times \text{Recall}}{\text{Precision} + \text{Recall}} \tag{8}$$

In order to better reflect the benefits of using parameters in each network model, we also defined a new evaluation index: parameter benefit (PB). This index was calculated as follows:

$$PB = \frac{Accuracy - Accuracy\ Threshold}{number\ of\ parameters} \tag{9}$$

Since the parameter benefit was derived from the high-precision network analysis, a PB value of less than zero meant that the network did not have high prediction accuracy. Therefore, it was necessary to decide whether to include the network with a PB value of less than zero (according to the actual situation) in the evaluation system. Thus, we would recommend that readers flexibly adjust the accuracy threshold (89.57% in this article) according to their own needs.

## 3. Data and Experiments

### 3.1. Dataset

Considering the accuracy and efficiency of label production, we used the GID dataset produced by the Wuhan University team in this experiment. This dataset consists of 150 GF-2 remote sensing images with five-category label maps. The size of each image is unified to 6800 × 7200. The images are distributed among 60 cities in China, covering an area of over 50,000 km$^2$. More detailed information on remote sensing imagery can be found in Table 1. Its large number of samples makes the dataset highly representative, and it is anticipated that the weight coefficients trained using the data set would be more adaptive. We screened out 50 remote sensing images with a uniform distribution of water systems and obtained a two-class label map of water system distribution after color-changing the label images. Subsequently, 40 of the images were used for training to achieve a ratio of 4:1. The remaining 10 sheets were used for prediction. Since the computer's memory could not accommodate the input of an entire remote sensing image at one time, it is also needed to randomly crop the 40 remote sensing images used for training, resulting in 20,000 256 × 256 small-sized images that were sent to the network for training. We also set the judgement conditions during random cropping to ensure that the proportion of water systems in each cropped image was not less than 10%. This step is aimed at alleviating the impact of uneven sample distribution on training to a certain extent while strengthening the network's learning of water systems at different scales.

**Table 1.** Detailed information of GF-2 multispectral imagery.

| Satellite Parameters | GF-2 Multispectral Imagery |
|:---:|:---:|
| Product level | 1A |
| Number of bands | 4 |
| Wavelength (μm) | Blue (0.42–0.52) |
| | Green (0.52–0.59) |
| | Red (0.63–0.69) |
| | Near-infrared (0.77–0.89) |
| Size | 6800 × 7200 |
| Spatial resolution (m) | 0.8 m pan/3.24 m MS |

### 3.2. Implementation Details

For the experimental platform, we used a graphics card with Nvidia GeForce RTX 2080 Ti 11 GB video memory. The software environment used was the Windows10 Professional 64-bit operating system. Additionally, the programming language used was Python, while the deep learning framework used was Keras, and the TensorFlow framework as the backend was selected as the tool to build the model. Lastly, the GPU computing platforms CUDA10.0 and cuDNN7.4 were used as the deep learning GPU acceleration library.

### 3.3. Extraction Results of BU-Net

The visualization results are shown in Figure 3. It can be seen that the BU-Net performed very well on the 10 test images; such water bodies as a slender river, a large area of continuous lakes or even islands, reefs and boat branches in the water body were accurately identified and distinguished. Accuracy ranged from as high as 99.52% to as low as 96.78%. The F1-score ranged between as high as 99.73% and as low as 97.91%. IoU was as high as 99.54% and as low as 95.90%. In terms of overall average values, Accuracy was 98.31% on average, while the F1-score averaged 98.89%, and the average IoU was 97.81%; see Table 2 for more detailed evaluation data of BU-Net. Of course, a comparison with the label map reveals the persistence of many problems, such as many isolated pixels that are misclassified as a whole in the image. In addition, the model tended to confuse the three pixels in the area near buildings and vegetation and water, and some slender rivers were identified as discontinuous and disconnected (see Figure 4). Nevertheless, overall, the BU-Net network structure offered high prediction accuracy.

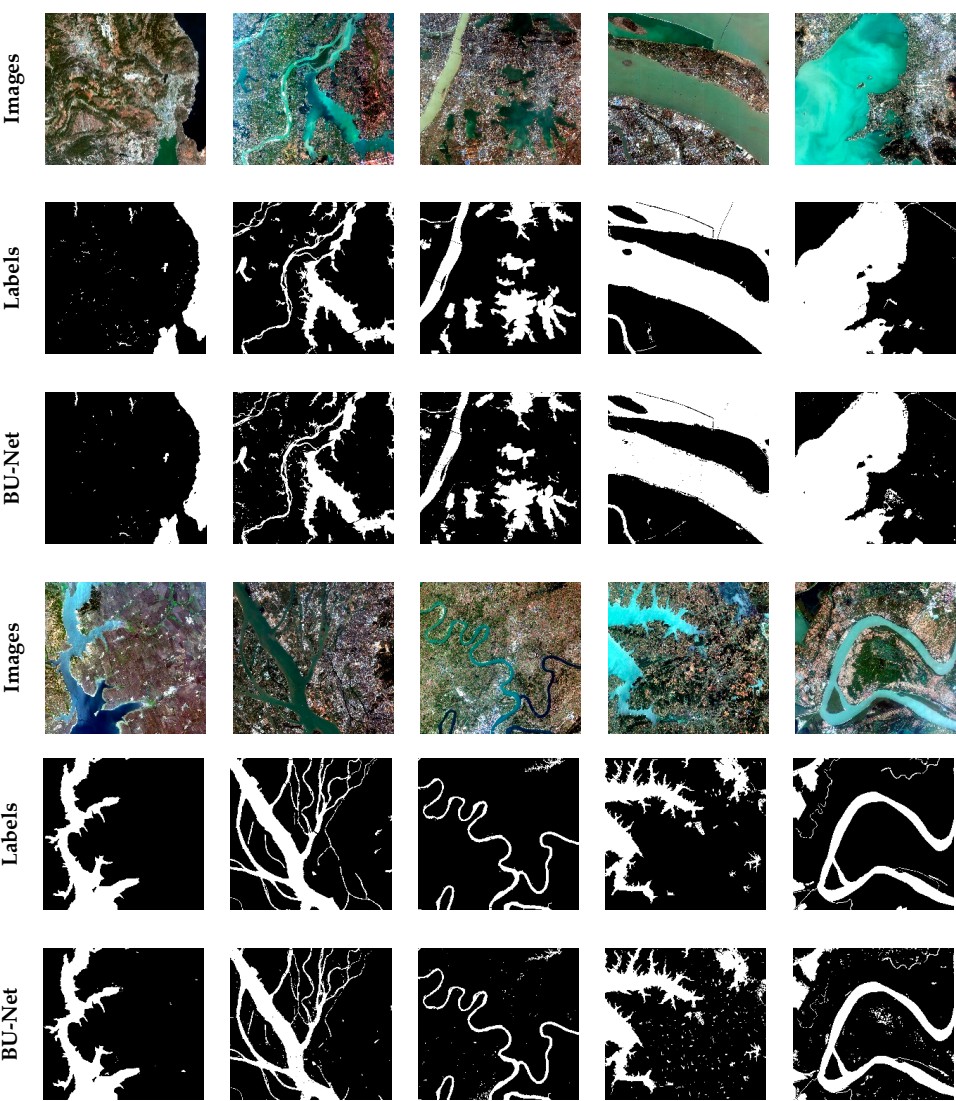

**Figure 3.** Prediction results of ten validation images.

**Table 2.** Results of OA, F1-score and IoU on the validation images.

| Test Images | OA (%) | F1-Score (%) | IoU (%) |
|---|---|---|---|
| All images | 98.31 | 98.89 | 97.81 |
| Image1 | 99.52 | 99.73 | 99.46 |
| Image2 | 97.89 | 98.71 | 97.45 |
| Image3 | 97.51 | 98.39 | 96.83 |
| Image4 | 98.00 | 97.91 | 95.90 |
| Image5 | 98.93 | 98.93 | 97.89 |
| Image6 | 99.62 | 99.77 | 99.54 |
| Image7 | 97.72 | 98.56 | 97.15 |
| Image8 | 99.23 | 99.59 | 99.17 |
| Image9 | 97.93 | 98.69 | 97.42 |
| Image10 | 96.78 | 98.00 | 96.07 |

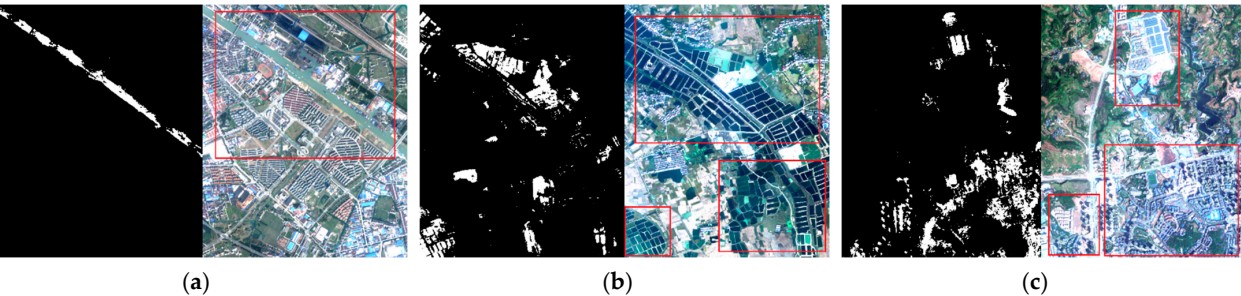

(**a**)　　　　　　　　　(**b**)　　　　　　　　　(**c**)

**Figure 4.** Remote sensing images misclassified by BU-Net: (**a**) Slender rivers that are identified as discontinuous; (**b**) paddy fields that are misclassified as water bodies; and (**c**) industrial and residential land that are misclassified as water bodies.

### 3.4. Comparative Experiment Using Different Parameters

For the verification of the semantic gap discussed above and to determine the most appropriate magnification and number of base convolution kernels for water extraction, eight combinations of the magnification and base convolution kernel number were tested. Combining the prediction indicators (see Table 3 and Figure 5), the prediction effect of BU-Net with a magnification of 2 was better than that of BU-Net with a magnification of 4, indicating that the semantic gap posed a notable problem, affecting the accuracy of the entire network structure for classification tasks. However, under the same magnification, the prediction effect of BU-Net with a small number of basic convolution kernels was better than that of BU-Net with a large number of basic convolution kernels, proving to a certain extent that increasing the number of network parameters could not necessarily yield better results. Through comparative analysis, we determined a satisfactory combination of magnification and the number of basic convolution kernels where the magnification was 2 and the number of basic convolution kernels was 32. Hence, in all experiments conducted in this paper, all unspecified BU-Nets were structures with a magnification of 2 and number of basic convolution kernels of 32.

**Table 3.** Results of different parameters.

| Methods | OA (%) | F1-Score (%) | IoU (%) |
|---|---|---|---|
| BU-Net (2.16) | 90.21 | 93.92 | 88.54 |
| BU-Net (2.32) | 98.31 | 98.89 | 97.81 |
| BU-Net (2.64) | 96.14 | 97.46 | 95.05 |
| BU-Net (2.128) | 93.55 | 95.91 | 92.14 |
| BU-Net (4.16) | 92.71 | 95.40 | 91.21 |
| BU-Net (4.32) | 98.18 | 98.81 | 97.64 |
| BU-Net (4.64) | 96.10 | 97.48 | 95.09 |
| BU-Net (4.128) | 96.38 | 97.67 | 95.45 |

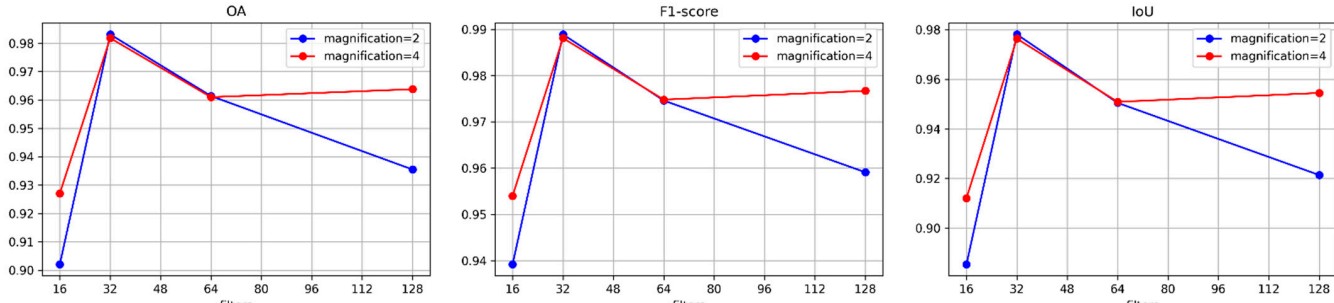

**Figure 5.** Results of different parameters.

### 3.5. Comparative Experiment of Different Networks

In order to more comprehensively analyze and evaluate the performance of BU-Net in water body recognition, five network structures (U-Net, SegNet, ResNet, DenseNet and PSPNet) and NDWI method were built to compare their classification effects (see Figure 6 and Table 4). U-Net and DenseNet showed the poorest performance of all of the networks. They could only identify some obvious water bodies and lacked the ability to identify water bodies having a dark color, turbid water quality or low contrast with adjacent areas. In comparison, while PSPNet and SegNet could identify the general outline of the water body in the image, finer details, such as small rivers and water body outlines, were still relatively vague. ResNet and BU-Net performed very well in terms of water body recognition. There was almost no difference between the predicted image and the label image, but some buildings and vegetation pixels similar to water bodies were still misclassified, resulting in many isolated pixels. Furthermore, this phenomenon was more obvious on BU-Net. In the future, we hope to solve this problem from the perspective of image morphology and further improve the prediction accuracy of BU-Net.

Then, the new metric: PB, defined above, together with prediction time, was applied as the evaluation criterion, in order to uncover useful information that can facilitate the continuous improvement and optimization of the network. As can be seen from Table 5, although ResNet and SegNet displayed high prediction accuracy, the huge number of parameters led to their low parameter utilization rate. In practical terms, the cost of training such a model with such a large number of parameters in a real-world setting is difficult to estimate. Meanwhile, DenseNet controlled the size of the model very well due to its unique cross-channel connection operation, which reduced a huge amount of parameter cost while increasing the computational space. Therefore, for a user who wants to train a deep-level DenseNet model, the hardware investment will be considerable, which limits the popular application of DenseNet. Although BU-Net fell short of ResNet in terms of prediction effect and was not as good as DenseNet Mini concerning model size, it effectively solves the defects of both in real-world applications and its extremely high parameter benefits tend to make up for the impact of insufficient parameters on prediction accuracy. In sum, BU-Net offers high practical value and significance.

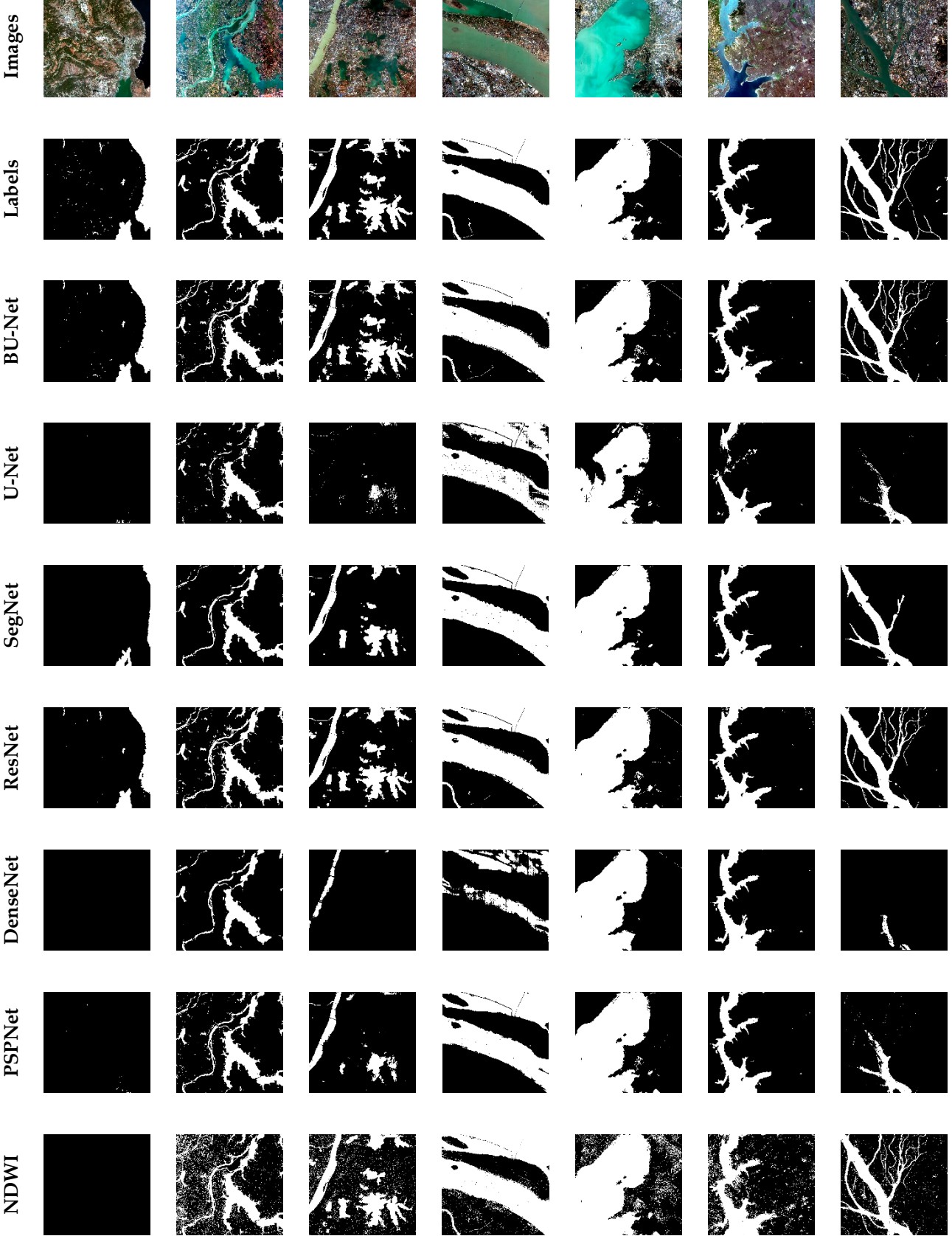

**Figure 6.** Prediction results of different networks.

**Table 4.** OA, F1-score and IoU of different networks.

| Methods | OA (%) | F1-Score (%) | IoU (%) |
|---|---|---|---|
| BU-Net | 98.31 | 98.89 | 97.81 |
| U-Net | 91.00 | 94.40 | 89.39 |
| SegNet | 95.69 | 97.24 | 94.63 |
| ResNet | 98.29 | 98.88 | 97.78 |
| DenseNet | 89.57 | 93.58 | 87.93 |
| PSPNet | 94.01 | 96.19 | 92.67 |
| NDWI | 89.04 | 92.44 | 85.94 |

**Table 5.** PB values of different networks.

| Methods | OA (%) | Number of Parameters (MB) | PB * | Prediction Time (s) |
|---|---|---|---|---|
| BU-Net | 98.31 | 33.3 | 0.2625 | 41 |
| U-Net | 91.00 | 355.0 | 0.0040 | 67 |
| SegNet | 95.69 | 364.0 | 0.0168 | 65 |
| ResNet | 98.29 | 377.0 | 0.0231 | 64 |
| DenseNet | 89.57 | 15.9 | 0.0000 | 934 |
| PSPNet | 94.01 | 45.9 | 0.0967 | 53 |

* The accuracy threshold is 89.57% in this article.

## 4. Discussion

In this article, the BU-Net method for water extraction was proposed. Compared with the traditional water body extraction method, this method demonstrated higher accuracy and a smaller model proportion, improving the application of automatic water body extraction from remote sensing images.

Although the BU-Net can basically meet the accuracy needs of current water extraction tasks, some questions remain about its structure. The residual structure features the bottleneck structure, which was originally designed to solve the problem of gradient disappearance and explosion during the training process of the network while increasing the amount of computation and the use of parameters. Although the effect of this structure is significant in terms of using deep networks for multi-classification problems, it is still a matter of debate whether the existence of lightweight networks for binary classification tasks features more advantages than disadvantages or more disadvantages. That said, readers can choose according to the needs of classification tasks.

In addition, it can be seen from BU-Net's identification results that the network is more inclined to identify water bodies from the perspective of the spectrum and does not effectively use the image's contour information. This factor highlights the need to further improve the network structure of BU-Net. In the future, we will consider alternative networks related to image recognition.

## 5. Conclusions

This paper addressed the problems of how to further improve the extraction effect of water bodies in high-resolution remote sensing imagery and the potentially oversized structure of deep convolutional networks. Accordingly, we proposed a fully convolutional neural network, BU-Net, for water body extraction, based on encoder–decoder architecture, and tested it using the GID dataset. The experimental results revealed that the model obtained water body extraction results with higher OA, F1-score and IoU values and greatly reduced the size of the model and prediction time while ensuring the extraction effect. The effectiveness of the BU-Net means that it is helpful for the rapid acquisition of water body information. This investigation also provides a new way of thinking for the architecture of water body extraction networks for remote sensing imagery.

**Author Contributions:** Writing—review & editing, S.A. and X.R. All authors have read and agreed to the published version of the manuscript.

**Funding:** This research was funded by the National Key Research and Development Program of China (Grant No. 2019YFC1804304), the National Natural Science Foundation of China (Grant No. 41771478), and the Fundamental Research Funds for the Central Universities (Grant No. 2019B02514).

**Data Availability Statement:** Not applicable.

**Conflicts of Interest:** The authors declare no conflict of interest.

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
