# Peer review of "A High-Precision Water Body Extraction Method Based on Improved Lightweight U-Net"

_remotesensing, doi:10.3390/rs14174127_

Round 1

Reviewer 1 Report

The assumption that the extraction effect of a convolutional neural network does not depend on the number of its parameters and reasonable adjustment of the network structure according to the characteristics of water bodies on remote sensing imagery can effectively improve the recognition effect is very interesting and a worthy approach to be testified; in my opinion, this is the key message and focus of the paper. Nevertheless, there are some confusing sentences, wrongly chosen phrases, inconsistent terminology. The use of language is good but can be convoluted at timed, I would suggest the authors fix these errors. The idea and main message of the paper are fine, but the paper must be improved before publication.

Comments:

1. The main methods applied are described in too much detail in the abstract. I suggest that they should be shortened.

2. Page 4, Line 6 and 7: These two numbers appearing in the sentence should be explained using a simple calculation.

3. Page 7, Line 18-21: It is best to use some suitable pictures to illustrate the extraction defects of the BU-Net network.

4. Page 8: Table 3 does not intuitively see the changes of BU-Net under different parameters. I recommend using the form of a statistical graph.

5. Table4 and table5 have the same header content, they can be replaced with more specific content to highlight their differences.

6. I suggest stating the Accuracy Threshold used by this table in the footer of table 5 to avoid misunderstanding.

7. Page 10, Line 28-29: The sentence about “the problems of poor water body recognition in high-resolution remote sensing images” is inaccurate, because the introduction mentioned that many existing methods have achieved good accuracy.

8. Page1

Line4 (left): has->have

9. Page 2

Line 10 (right): , and->and

Line 53 (left): I suggest using “we aim” instead of “aiming”

10. Page 3

Line 39 (left): classic->classical

11. Page 6

Line 37 (left): the table 1-> table 1

Author Response

Dear reviewer,

Thank you for the opinions, these comments are very helpful to improve the quality of the manuscript. We carefully revised our manuscript, further clarify the logic of writing for improve the quality of the manuscript.  Now I response the reviewer's comments with a point by point and highlight the changes in revised manuscript. Full details of the files are listed. We sincerely hope that you find our responses and modifications satisfactory and that the manuscript is now acceptable for publication.

Comment 1: The main methods applied are described in too much detail in the abstract. I suggest that they should be shortened.

Response: We are very sorry for the long abstract and it was rectified in Line 7-21.

Comment 2: Page 4, Line 6 and 7: These two numbers appearing in the sentence should be explained using a simple calculation.

Response: We are very sorry for our careless mistake and it was rectified at Line 156 and Line 157.

Comment 3: Page 7, Line 18-21: It is best to use some suitable pictures to illustrate the extraction defects of the BU-Net network.

Response: We agree that adding pictures would be useful to clarify the problem and the new picture was placed at Line 295.

Comment 4: Page 8: Table 3 does not intuitively see the changes of BU-Net under different parameters. I recommend using the form of a statistical graph.

Response: We agree that using statistical graph would be useful to observe changes in data and the adding picture was placed at Line 315.

Comment 5: Table 4 and Table 5 have the same header content, they can be replaced with more specific content to highlight their differences.

Response: We are very sorry for our careless mistake and it was rectified at Line 332 and Line 350.

Comment 6: I suggest stating the Accuracy Threshold used by this table in the footer of table 5 to avoid misunderstanding.

Response: We are very sorry for our careless mistake and it was rectified at Line 351.

Comment 7: Page 10, Line 28-29: The sentence about “the problems of poor water body recognition in high-resolution remote sensing images” is inaccurate, because the introduction mentioned that many existing methods have achieved good accuracy.

Response: We are sorry for our inappropriate elaboration and it was rectified in Line 372-381.

Comment 8-11: 

Page1

Line4 (left): has->have

Page 2

Line 10 (right): , and->and

Line 53 (left): I suggest using “we aim” instead of “aiming”

Page 3

Line 39 (left): classic->classical

Page 6

Line 37 (left): the table 1-> table 1

Response: We are very sorry for our careless mistake and it was rectified at Line 45,Line 91,Line 138 and Line 249.

Reviewer 2 Report

General comments:

It's meaningful for water resource management and disaster assessment to study the temporal distribution laws. The authors of this manuscript proposed a more lightweight network based on the high-precision water body extraction neural network, which is very helpful for the rapid acquisition of water body information. This manuscript is fluent in language, and meet to the writing norms of academic papers. Although there are some problems and mistakes which are need to be deal with, I suggest that this manuscript can be accepted after minor revision.

Specific comments are listed below:

1. The abstract needs to be shortened to keep around 200 words.

2. The introduction mentions that the NDWI method has great problems in the extraction using remote sensing images, but the extraction effect of the NDWI method was not tested in the comparative experiments.

3. The introduction spends a lot of space to write ideas for the improvement of network, but the significance of the improvement is not very obvious. There should be more words to describe the innovative nature of this network.

4. Section 2 and Section 3 have the same title, which should be a writing error.

5. The header of the first column of table 4 does not correspond to the content.

6. The formulas appearing in the manuscript are not numbered.

7. The calculation formulas of OA and F1-Score used in the accuracy evaluation are not mentioned.

8. Page 1

  Line 4 (left): has->have.

9. Page 2

  Line 5 (right)(and further): check the usage of commas and especially together with “and”.

  Line 11 (left): in general->in generally.

10. Page 3

   Line 39 (left): classic->classical.

11. Page 6

   Line 37 (left): in the table 1->in table 1.

Author Response

Dear reviewer,

Thank you for the opinions, these comments are very helpful to improve the quality of the manuscript. We carefully revised our manuscript, further clarify the logic of writing for improve the quality of the manuscript.  Now I response the reviewer's comments with a point by point and highlight the changes in revised manuscript. Full details of the files are listed. We sincerely hope that you find our responses and modifications satisfactory and that the manuscript is now acceptable for publication.

Comment 1: The abstract needs to be shortened to keep around 200 words.

Response: We are very sorry for the long abstract and it was rectified in Line 7-21.

Comment 2: The introduction mentions that the NDWI method has great problems in the extraction using remote sensing images, but the extraction effect of the NDWI method was not tested in the comparative experiments.

Response:  We agree that adding data would be useful to articulate the facts and it was added at Line 332.

Comment 3: The introduction spends a lot of space to write ideas for the improvement of network, but the significance of the improvement is not very obvious. There should be more words to describe the innovative nature of this network.

Response: We are sorry for our inappropriate elaboration and it was rectified in Line 74-81.

Comment 4: Section 2 and Section 3 have the same title, which should be a writing error.

Response: We are very sorry for our careless mistake and it was rectified at Line 242.

Comment 5: The header of the first column of table 4 does not correspond to the content.

Response: We are very sorry for our careless mistake and it was rectified at Line 332.

Comment 6: The formulas appearing in the manuscript are not numbered.

Response: We are very sorry for our careless mistake and it was rectified at Line 232.

Comment 7: The calculation formulas of OA and F1-Score used in the accuracy evaluation are not mentioned.

Response: We are very sorry for our careless mistake and it was rectified at Line 232.

Comment 8-11: 

Page 1

Line 4 (left): has->have.

Page 2

Line 5 (right)(and further): check the usage of commas and especially together with “and”.

Line 11 (left): in general->in generally.

Page 3

Line 39 (left): classic->classical.

Page 6

Line 37 (left): in the table 1->in table 1.

Response: We are very sorry for our careless mistake and it was rectified at Line 40,Line 45,Line 45,Line 138 and Line 249.

Reviewer 3 Report

This manuscript studied water body extraction from remote sensing images based on deep learning. In my opinion, the pursuit of lightweight convolutional neural networks is very promising. The language of the manuscript is fluent and the idea is novel. But there are some problems, which must be solved before it is considered for publication. If the following problems are well-addressed, I believe that the essential contribution of this paper are important for the popularization and utilization of remote sensing information.

Comments:

1. The abstract mentions that the method in this manuscript not only improves the accuracy, but also reduces the training time. However, the experimental part does not mention the change in the training time after the improvement.

2. The introduction lists a lot of methods for deep learning to extract water bodies, but the advantages of the method proposed in this manuscript compared to some current mainstream networks are not very clear.

3. Regarding the classification effect of BU-Net under different parameters, the manuscript only makes five combinations. The data samples are too few.

4. When using English abbreviations for the first time in the manuscript, you should put the abbreviation in parentheses instead of the full name.

5. “Material and Methods” are generally placed in the second section. Combined with the content of the third section, I think the title of the third chapter should be changed to “Data and Experiment” or other similar expressions.

6. The full text appears “in order to” many times, please consider using other synonyms instead.

7. Page 1, Line 4 (left): has->have.

Line 14 (right): adapt->adapt to.

8. Page 2, Line 5 (right): delete comma.

Line 10 (right): delete comma.

Line 11 (left): general->generally.

9. Page 3, Line 39 (left): classic->classical.

10. Page 6, Line 6 (left): wrong punctuation.

Author Response

Dear reviewer,

Thank you for the opinions, these comments are very helpful to improve the quality of the manuscript. We carefully revised our manuscript, further clarify the logic of writing for improve the quality of the manuscript.  Now I response the reviewer's comments with a point by point and highlight the changes in revised manuscript. Full details of the files are listed. We sincerely hope that you find our responses and modifications satisfactory and that the manuscript is now acceptable for publication.

Comment 1: The abstract mentions that the method in this manuscript not only improves the accuracy, but also reduces the training time. However, the experimental part does not mention the change in the training time after the improvement.

Response: We agree that adding data would be useful to clarify the problem and the  data was added at Line 350.

Comment 2: The introduction lists a lot of methods for deep learning to extract water bodies, but the advantages of the method proposed in this manuscript compared to some current mainstream networks are not very clear.

Response:  We are sorry for our inappropriate elaboration and it was rectified in Line 74-81.

Comment 3: Regarding the classification effect of BU-Net under different parameters, the manuscript only makes five combinations. The data samples are too few.

Response: We agree that adding data would be useful to clarify the problem and the  data was added at Line 314.

Comment 4: When using English abbreviations for the first time in the manuscript, you should put the abbreviation in parentheses instead of the full name.

Response: We are very sorry for our careless mistake and it was rectified in Line 215-230.

Comment 5: “Material and Methods” are generally placed in the second section. Combined with the content of the third section, I think the title of the third chapter should be changed to “Data and Experiment” or other similar expressions.

Response: We are very sorry for our careless mistake and it was rectified at Line 242.

Comment 6: The full text appears “in order to” many times, please consider using other synonyms instead.

Response: We are very sorry for our careless mistake and it was rectified at Line 187,Line 197,Line 213,Line 244 and Line 299.

Comment 7-10: 

Page 1, Line 4 (left): has->have.

Line 14 (right): adapt->adapt to.

Page 2, Line 5 (right): delete comma.

Line 10 (right): delete comma.

Line 11 (left): general->generally.

Page 3, Line 39 (left): classic->classical.

Page 6, Line 6 (left): wrong punctuation.

Response: We are very sorry for our careless mistake and it was rectified at Line 16,Line 40,Line 45,Line 45,Line 138 and Line 220.

This manuscript is a resubmission of an earlier submission. The following is a list of the peer review reports and author responses from that submission.